# Contrasting Dynamics of Littoral and Riparian Reed Stands within a Wetland Complex of Lake Cerknica

**DOI:** 10.3390/plants12051006

**Published:** 2023-02-22

**Authors:** Nik Ojdanič, Igor Zelnik, Matej Holcar, Alenka Gaberščik, Aleksandra Golob

**Affiliations:** Department of Biology, Biotechnical Faculty, University of Ljubljana, Jamnikarjeva 101, 1000 Ljubljana, Slovenia

**Keywords:** *Phragmites australis*, remote sensing, intermittent wetland, NDVI, plant productivity, phenology, littoral, riparian, Sentinel-2, plant biomass

## Abstract

This contribution discusses the use of field measurements and remotely sensed data in an exploration of the effects of environmental parameters on the riparian and littoral stands of the common reed (*Phragmites australis*) in an intermittent wetland in Slovenia. For this purpose, we created a normalized difference vegetation index (NDVI) time series extending from 2017 to 2021. Data were collected and fitted to a unimodal growth model, from which we determined three different stages relating to the reed’s growth. The field data consisted of the above-ground biomass harvested at the end of the vegetation season. Maximal NDVI values at the peak of the growing season exhibited no useful relationship with the above-ground biomass at the end of the season. Intense and long-lasting floods, especially during the period of intense culm growth, hindered the production of common reeds, while dry periods and temperatures were helpful before reed growth began. Summer droughts exhibited little effect. Water level fluctuations exerted a greater effect on reeds at the littoral site due to more pronounced extremes. In contrast, more constant and moderate conditions at the riparian site benefited the growth and productivity of the common reed. These results can prove useful for decision making regarding common reed management at the intermittent lake Cerknica.

## 1. Introduction

The common reed (*Phragmites australis* (Cav.) Trin. ex Steud) grows in a wide range of habitats, including the riparian zones and littoral zones of lakes. Its ability to thrive in different habitats with diverse environmental conditions can be attributed to its high phenotypic plasticity [1] and genetic diversity [2]. The common reed is one of the most common species in wetlands and is considered to be highly productive and a significant contributor to the total biomass of a wetland [3]. Thus, reed biomass estimation is important because biomass production reflects the stability and productivity of wetlands [4]. Water level changes in time and space play an important role in the vitality and productivity of reeds, as they largely define the conditions in the reeds’ habitat [5,6]. The common reed is adapted to sustain floods of certain depths as it contains aerenchyma channels that provide oxygen to submerged organs and the rhizosphere. The gas-space system associated with the reed provides pathways for pressurized convective flows of atmospheric gases [7,8] by Venturi suction which is caused by the wind blowing across the tops of dead culms and by humidity-induced Knudsen diffusion, which is initiated in living sheaths and culm nodes [9,10]. Flooding events may reduce convective gas flow to the basal parts of reeds [11], which, in turn, leads to oxygen deficiency in the roots and rhizomes, thus changing the metabolic state of the whole plant [12]. Flooding events may also lead to denitrification because of anoxic conditions in the soil [13], which, in turn, may lower the amount of nitrate available to the plants. All these processes affect reed growth and development, and thus its production.

Most field-based methods for the determination of biomass, though accurate and reliable, are time-consuming, destructive, and often limited to smaller areas. Remotely sensed data present an economically efficient alternative that also allows greater spatial coverage and is substantially less time-consuming [14]. Vegetation indices based on multispectral data can be used to predict plant biomass [15]: a practice that is commonly used in wetlands. In addition, remote sensing techniques can be utilized to map the phenology of wetlands [16]. Remote-sensing techniques based on vegetation indices are supported by field data and may prove to be a useful tool for long-term monitoring as has been shown for seasonal marsh ecosystems [17]. These methods have also been applied to studies regarding the common reed and, as a result, Sentinel-2 data have been shown to be the most accurate when compared to other widely used multispectral satellite data sources, namely Landsat-7 and Landsat-8 [18]. One of the most commonly used indices in remote sensing is the normalized difference vegetation index (NDVI) [19,20]. NDVI has various uses, including biomass estimation [21], plant productivity monitoring [22], plant stress detection [23], and leaf water potential estimation [24]. NDVI values tend to increase as the plants develop, and NDVI time series data may, therefore, also be used for the monitoring of plant phenology [25,26,27].

Such approaches for plant monitoring are especially useful in wetlands where the water regime often prevents the use of field methods, such as harvesting. This is also the case in the intermittent wetland lake Cerknica, located in southwestern Slovenia. The defining characteristic of intermittent wetlands is fluctuating water levels with distinct wet and dry periods [28]. Abundant rainfall in the catchment area of the Cerknica lake during autumn and spring causes the area to be flooded on average for 260 days in a year [29]. The dry period usually occurs during summer. These water level fluctuations influence the energy flow and the turnover of matter within the lake, greatly impacting the function of the entire ecosystem [30]. The water regime is also known to be a key driver of the development and zonation of wetland plant communities [31,32]. The predominant plant species of the intermittent lake Cerknica is the common reed [33], which exhibits a highly variable biomass production across years [5]. According to Lumbierres et al. [17], the knowledge of the spatiotemporal pattern of biomass production is important, especially for the management of wetlands with variable flooding regimes.

The aim of this current study was to introduce remotely sensed data into research at lake Cerknica as remote sensing may offer valuable insights regarding the productivity and phenology of the common reed in this unique ecosystem. For this purpose, we examined a possible correlation between NDVI data and the above-ground biomass (AGB) in stands of common reeds with the goal of optimizing the monitoring of the annual productivity of reed stands. We also fitted a 5-year NDVI time series to a unimodal growth model, which gave us an opportunity to explore the effects of temperature and water level on the common reed at different growth stages. These results may be applicable to future management decisions which affect the common reed at lake Cerknica. We hypothesized that maximal NDVI values from our fitted data could be used in determining the AGB at the end of the vegetation season. We also postulated that water level fluctuations should primarily impact the productivity of common reeds depending on the stage of their development, especially in the littoral reed stand.

## 2. Materials and Methods

The workflow in the examination of the effects of environmental variables in different growth periods and the comparison of biomass and remotely sensed parameters is shown in Figure 1. This includes tasks related to satellite images, field harvesting of the biomass of common reeds, and the collection of environmental data, specifically data on the water level and temperature. Remotely sensed data, in combination with a growth model, permits the determination of growth stages in the common reed. This, in turn, allows us to test how mean temperatures, maximal temperatures, and water levels affect the productivity of the common reed in two different reed stands. 

### 2.1. Study Area

The observed areas containing the stands with a dominant common reed are located at lake Cerknica: an intermittent lake in southwestern Slovenia. The area of the lake is of karst origin, mainly consisting of Mesozoic limestone and dolomite [34]. The majority of the inflows that fill the lake are from the south-eastern and eastern edge of a karstic valley named Cerkniško Polje [35]. High discharges of inflowing watercourses to these karstic features cause seasonal floods on poljes. Drying is the result of water escaping through numerous sinkholes. We observed two reed stands growing in different abiotic conditions. The first of these was a riparian reed stand, which is located on the bank of the river Stržen, and the second is a littoral reed stand, more than 1 km away from the river Stržen (Figure 2) on the gently sloping bottom of lake Cerknica. The water quality of both areas is comparable and considered non-eutrophic when the lake is filled with water [36,37].

### 2.2. Environmental Variables

All the environmental variables regarding water levels and temperatures were obtained from the Slovenian Environmental Agency (https://www.arso.gov.si (accessed on 10 October 2022)). Water levels were obtained from the two nearby water level measuring stations alongside the river Stržen. The water level measuring station at Gorenje Jezero best represents the water level at the riparian reed stand, while the water level measuring station at Dolenje Jezero best represents the water level at the littoral reed stand in Zadnji kraj (Table 1, Figure 2). We used a GNSS antenna (Ardusimple, Andorra) to determine the exact altitude of each of the water level measuring stations. Information concerning the average daily and maximal daily temperatures was obtained from the nearest weather measuring station (13.8 and 15 km air distance) located in Postojna (Table 1).

### 2.3. Satellite Imagery and NDVI Dataset

For the remote sensing in this study, we used images from the Sentinel-2 database (https://scihub.copernicus.eu/dhus/#/home (accessed on 16 April 2022)). Sentinel-2 has a revisit time of 5 days, and its imagery is publicly available from the European Space Agency. We collected all the available images from the start of the growing season in April to the end of the growing season in September for each year from 2017 to 2021. Atmospherically corrected Level-2A images were used when calculating NDVI. Level-1C images were tested for cloud interference by creating cloud masks using the ESA SNAP software (version 9.0.0). Images where the generated cloud mask layers appeared over the observed reed sites were then removed. At each observed site, we selected four adjacent pixels in which the common reed was the prevailing species (the abundance of common reed at each pixel was verified on site). Each pixel had a spatial resolution of 10 m. For each pixel, we then calculated the NDVI using the following equation:*NDVI* = (*NIR* − *Red*)/(*NIR* + *Red*)(1)
where *NIR* represents the reflectance at 832.8 nm in the near-infrared region of the electromagnetic spectrum and *Red* represents the reflectance at 664.6 nm in the red region of the electromagnetic spectrum.

### 2.4. NDVI Growth Model

After the completion of our NDVI dataset, we filtered out all the data after the NDVI reached its peak values, thus eliminating the senescence part of the phenology. For each year and location, we then fitted our data to a model that resulted in the best fit. This model depicted the unimodal trajectory of plant biomass accumulation and was calculated as suggested by Tóth [26]: *y* = *y*_0_ + *a*(1 − *e*^−*bx*^)(2)
where *y*_0_ represents the initial NDVI value from the Sentinel-2 dataset for each year and location and *x* represents the date on which the NDVI values appear. Data were fitted to the model using the *nls* function from the stats (v3.6.2) package, which is part of base R. acquired equations and statistics, which are presented in Appendix A. Due to large gaps in the missing data for the littoral reed stand at the beginning of the 2019 and 2020 seasons, 51- and 25-day transformations were applied, respectively, for the date of the initial NDVI value in order to acquire significant fits. Transformations were applied because, without them, the date of the initial NDVI value was underestimated. Based on the model, we then calculated three additional parameters. The first parameter was the maximal seasonal NDVI, which represented the predicted maximal value from each fit, and was calculated as follows [26]: *NDVI*_*max*_ = *y*_0_ + *a*(3)
where *y*_0_ represents the first NDVI value from the Sentinel-2 dataset for each year and location, and *a* represents a constant within the model. The second parameter is the time of maximum NDVI intensity increase which is based on the slope of the model and represents the date with the most intense growth. It was acquired using the following equation [26]:*GR*_*max*_ = (1 − *ln*(*a* − *y*_0_/*a*))/*b*(4)
where *a* and *b* represent constants within each fit. The final parameter is purely theoretical as it depicts the initial rate of NDVI increase and is calculated as follows [26]:*α* = *NDVI*_*max*_/*GR*_*max*_(5)

### 2.5. Biomass at the End of the Reproductive Phase

Each year in September, biomass was harvested from four 0.5 × 0.5 m plots at each location. The size of the plots was determined, as suggested by Gaberščik et al. [38]. The above-ground biomass (AGB) of all the reed plants from each plot was harvested and dried to a constant mass, by which we obtained the average dry biomass of each plot. The AGB values were converted into grams/m^2^. The data regarding dry AGB were tested for correlations with our NDVI dataset using Spearman’s rank correlation coefficient. The Shapiro–Wilk normality test was conducted beforehand. We also used the GNSS antenna to determine the exact altitude of the littoral and riparian reed stands. 

### 2.6. Redundancy Analysis

To explore the relationship between model parameters, AGB, and environmental variables (water level and temperature data), we split our environmental variables into three different periods, based on the growing season, model, and NDVI dataset (Figure 3). The first period was classified as the period before the reeds’ establishment. It started at the beginning of the growing season (first of April) and lasted until our model hit the NDVI value of 0.2 (in our NDVI dataset, NDVI values started increasing from 0.2, thus marking this as the beginning value of reed growth). The second period was classified as the vegetative phase. It began at the 0.2 NDVI value mark and lasted until the NDVI value that increased was less than 10% between consecutive acquired NDVI’s. The last period was classified as the period of NDVI saturation. It began at the end of the second period and lasted until the end of our yearly NDVI dataset. 

For each period, we calculated the average daily temperature, average daily maximal temperature, the number of days in which the area was flooded, and the number of days in which the area was dry. The aforementioned altitudes of both water level measuring stations allowed us to transform the measurements from both the water level measuring stations into the exact elevation above sea level. This information, coupled with the altitudes of both reed stands, allowing us to determine whether our observed areas were flooded or dry. A redundancy analysis (RDA) was performed to reveal the significance of the environmental variables. A relationship was explored between a matrix of response variables containing AGB and model parameters (rNDVI_max_, mNDVI_max_, GR_max,_ and α) and a matrix of explanatory variables (number of flooded days, number of dry days, average daily temperature, and maximal daily temperature per growth period). To avoid collinearity, a forward selection of explanatory parameters was applied using Monte Carlo tests with 499 permutations. These analyses were performed using the Canoco 5 program (Microcomputer Power: Ithaca, NY, USA). 

## 3. Results

When comparing the results of Spearman’s rank-order correlation test, no strong correlations were found (Table 2). A weak statistical significance with a low correlation coefficient was detected when pairing AGB and maximal NDVI values acquired from the Sentinel-2 derived dataset. Pairing AGB with maximal modelled NDVI values was statistically insignificant. 

When comparing the results of applied fits to the model, differences could be seen both within each individual reed stand and also between the two sites during the 5-year period (Table 3). The date of growth starts in our fits showed that reeds at the riparian site appeared to enter a stage of rapid growth sooner than those in the littoral site. This is also supported by an increase in the time growth rate, which appeared sooner at the riparian reed stand. Modeled maximal NDVI values showed similar values as our real-time NDVI maximal values, apart from the year 2019 at the littoral site. However, a 51-day transformation of the initial NDVI date was needed in this instance due to a large gap of missing data at the beginning of the growing season. The AGB at the end of the growing season was highest at the riparian location in the year 2020. The years 2017 and 2018 were reported to have the lowest AGB. The AGB at the littoral reed stand was also higher in the years 2020 and 2021. In a comparison of the annual AGB between both locations, littoral reeds exhibited lower values. For the best understanding of differences between both reed stands, the modeled fits had to be paired with data regarding environmental variables: mainly the water level data. 

### 3.1. The 2017 Growing Season

In 2017 a clear difference between the riparian and littoral reed was is apparent. The riparian reed stands at Gorenje Jezero reached NDVI saturation before the littoral reed stand at Zadnji kraj (Figure 4a). The time of maximal growth for littoral reeds appeared 21 days later than their riparian counterpart, and the date of growth-start in the riparian stand occurred three days earlier than in the littoral stand. Once NDVI saturation occurred at both sites, the NDVI values did not differ further. During the growing season, the water level crossed the inundation line at both sites. In both instances, it occurred at the start of the growing season, but compared to the riparian site, the littoral reed stand remained flooded longer (Figure 4b and Appendix A). Before summer, average temperatures remained around 12 °C and rose in the summer months (Figure 4c and Appendix A). Maximal daily temperatures followed the same trend, and at the end of the growing season, the AGB was 227.4 g higher at the riparian site than at the littoral stand (Table 3).

### 3.2. The 2018 Growing Season

During the 2018 growing season, a similar observation regarding the growth rate was made. Growth at the riparian stand started two days before growth at the littoral stand, and the riparian reed stand reached NDVI saturation before the littoral reed stand (Figure 5a). The time of maximal growth at the littoral site appeared 11 days after its riparian counterpart. The date of growth started in the riparian reed stand and occurred two days before its littoral counterpart (Table 3). The maximal NDVI values after saturation also differed, as the NDVI was higher at the riparian stand. Flooding was more prominent at the littoral site. The area was flooded to a depth of over 1 m at the start of the growing season, and flooding lasted until the summer. In the summer months, the water level remained relatively high at the littoral site indicating that water shortage was not an issue in 2018 (Figure 5b and Appendix A). Average temperatures appeared to be constant through spring and summer (Figure 5c and Appendix A). At the end of the growing season, the AGB was again reported to be higher at the riparian site, with a 235.6 g difference in mass when compared to AGB at the littoral stand (Table 3).

### 3.3. The 2019 Growing Season

For the 2019 growing season, the most apparent difference at both sites was the date of growth start. The date of growth-start at the riparian site occurred on 12 April, while at the littoral site, it was on 5 June. The date of growth-start at the littoral site was hypothetical, as a 51-day transformation was needed due to a gap of missing data caused by cloud interference. Nevertheless, NDVI saturation again appeared to follow the same pattern (Figure 6a). The riparian reed stand reached NDVI saturation before the littoral stand. Differences in maximal NDVI values were not apparent. In this year 2019, flooding was more intense during May and reached a peak in June. The water level at the littoral reed stand was maximized at over 1.5 m, while the riparian site peaked at slightly below 1 m above the ground (Figure 6b and Appendix A). Spring average temperatures varied from 8 to 12 °C, whereas summer average temperatures were obviously higher (Figure 6c and Appendix A). Maximum daily temperatures followed the same trend. In 2019, a drastic difference in biomass was measured at the end of the growing season. For the littoral stand, AGB was reported at 215.3 g, while for the riparian stand, the AGB was 804.3 g (Table 3). 

### 3.4. The 2020 Growing Season

In the 2020 season, NDVI saturation again appeared faster at the riparian reed stand (Figure 7a). The date of growth start at the riparian site was 19 April. The date of the growth start at the littoral site was on 15 May. Maximal growth at the riparian site was reached 11 days after the start of the growth. For the littoral site, maximal growth appeared 18 days after growth start. Maximal NDVI values were lower at the littoral site compared to its riparian counterpart (Table 3). Flooding occurred in the first half of June at both locations. However, at the riparian site, flooding only lasted a few days, while at the littoral site, flooding persisted until the second half of July. The water level at the start of the growing season, before the flooding in summer, was low (Figure 7b and Appendix A). Spring average temperatures remained between 8 and 12 °C, while summer average temperatures were higher (Figure 7c and Appendix A). Maximal daily temperatures followed the same trend. AGB at the end of the season was 284.9 g higher at the riparian location (Table 3).

### 3.5. The 2021 Growing Season

In 2021, the riparian reed stand reached NDVI saturation before the littoral reed stand (Figure 8a). The date of the growth start at the riparian site was on 2 May. The date of the growth start of the littoral reed stand was on 30 May. Maximal NDVI values were slightly higher at the riparian site, where they reached 0.853, while the littoral site had a maximal NDVI value of 0.805 (Table 3). The riparian site was flooded in May, with a peak water level just above 0.5 m. The littoral reed stand was flooded from the start of the growing season until the second half of June, with a peak of almost 1.5 m above the ground in May. Initial flooding in April was less than 0.5 m (Figure 8b and Appendix A). Average temperatures in the spring months were notably lower than in the summer months (Figure 8c and Appendix A). The daily maximal temperatures followed the same trend. As in all previous years, AGB at the end of the season was 332.8 g higher at the riparian site than at the littoral site (Table 3). 

### 3.6. The Relationship between Environmental Parameters, Model Parameters and Above-Ground Biomass (AGB)

Results of the RDA showed that environmental parameters accounted for 84.2% of the variance in model parameters and AGB (Table 4). Variables regarding water level accounted for 62.5% of the total variance. The number of flooded days (FD) during the vegetative phase, NDVI saturation phase, and before the establishment phase explained was 28.6, 10.8, and 5.7% of the variance, respectively. The number of dry days (DD) was significant in the period of NDVI saturation and before the establishment phase, explaining 9.9 and 7.5% of the variance, respectively. The effect of the temperature was significant during the vegetative phase and before the establishment phase. Temperatures during the vegetative phase explain 13.8% of the variance as a whole, with average maximal temperatures (AvgT2_max_) explaining 8.8%. The temperatures before reed establishment began to account for a total of 11.6% of the variance, with average daily temperatures (AvgT1) explaining 7.4% of the variance.

An additional result derived from the RDA was an ordinate plot showing the relationship between model parameters and the AGB with environmental variables (Figure 9). A clear difference could be observed between the riparian site at Gorenje Jezero and the littoral site at Zadnji kraj. When compared to data from Zadnji kraj, the data from Gorenje Jezero appeared more consolidated across the 5-year period, while the data representing Zadnji kraj were much more diverse. A negative relationship between the vector of AGB and the number of flooded days during the vegetative stage (FD2) could be seen, as the vectors pointed in opposite directions. A similar but weaker relationship could be observed with the vectors of dry days before the establishment of the reeds (DD1) and AGB. The vectors regarding the temperatures before reed establishment (AvgT1 and AvgT1_max_) and during the vegetative phase (AvgT2 and AvgT2_max_) reveal a similar relationship with the vector of AGB. The vector of flooded days before reed establishment (FD1) is negatively related to the vector of initial NDVI growth. The vector of dry days during NDVI saturation (DD3) was positively related to vectors of maximal NDVI values. 

## 4. Discussion

Upon comparing the results of remotely sensed data with the results of the characteristics of reeds obtained in the field, no useful relationship could be established between our NDVI dataset and the AGB of reeds at the end of the season. This is probably due to the saturation phenomenon, in which NDVI becomes insensitive to changes when the biomass reaches a certain level [20]. The use of different vegetation indices, better spatial resolution, and regression models may yield better results [39,40,41]. However, the growth model based on our NDVI data allowed us to explore the effects of environmental conditions, mainly temperatures, and water level, in a novel manner at lake Cerknica. This model allowed us to estimate the start of the growing season of common reeds, as well as the period of intense growth of culms during the vegetative phase.

In this study, the water level was seen to play an important role in the productivity of reeds. Water level variables explained more than half of the variability, and most importantly, flooding events that occurred during the vegetative phase exerted a clear negative impact on the final AGB and NDVI values. Early occurrence of high-water levels hindered the early development of culms [42]. Additionally, extreme flooding events may have caused reed die-back syndrome [12]. Yi et al. [43] reported that high-water levels were most detrimental to the overall health of reeds when they simulated a suitable habitat. Models based on growth dynamics, altitude, and water levels revealed that high water levels in summer might be one of the most important factors in controlling the lakeside frontline of the reed [44]. The littoral reed stands at Zadnji kraj were affected by water level fluctuations more than the riparian stand at Gorenje jezero. Water is continually present at the riparian reed stand due to the proximity of the river Stržen but the stand was rarely flooded for longer periods due to its slightly elevated position, which presents a much more favorable habitat for common reeds [38]. At the littoral site, however, water level fluctuations are unpredictable and more extreme. This was most obvious in 2019 when floods persisted until the second half of June, and water levels reached above 1.5 m during the vegetative phase. In the same year, an AGB of only 215.3 g was reported in the littoral reed stand, which was 174.7 g less than the AGB in the previous year. In comparison, the riparian reed stand had a total AGB of 804.3 g in the same year. The intensity of growth was also greater at the riparian reed stand when compared to the littoral reed stand, indicating that extreme water level fluctuations also negatively impacted the growth process. It is also possible that flooding hinders mineralization processes due to a lack of oxygen [45], thus increasing the accumulation of toxic substances in the rhizosphere [46] and decreasing the availability of nutrients during reed growth. Maximal NDVI values also appeared to be affected by water level fluctuations, as NDVI values at the riparian reed stand were higher in the years 2018, 2020, and 2021. Deegan et al. [47] reported that water level fluctuations up to 30 cm did not affect reed biomass, but amplitudes of 45 cm were enough to hinder the production of common reeds. It is also possible that the loss of NO_3_^−^ due to denitrification may also affect reed culm growth as NO_3_^−^ is an essential macronutrient [48], although Chu et al. [49] reported that photosynthesis, metabolism, and common reed growth were maintained at high levels under N-deficient conditions indicating that common reed was well adapted to conditions of low nitrogen. Rickey and Anderson [50] contradicted this and reported that leaves yellowed and died back when the common reed was grown in an environment with low nitrogen.

Temperatures during the intense vegetative growth phase explained 13.8% of the total variance. A negative relationship with high temperatures was also observed in this period and was found to negatively affect the final AGB of reeds. High temperatures during the vegetative phase also appeared to lower the intensity of the NDVI increase. Physiological processes are often temperature-dependent, but our results contradict the findings regarding the effect of high temperatures on common reeds. Zemlin et al. [51] reported that morphological parameters show only slight relation in average temperatures during the growth phase, with shoot length appearing longer at higher temperatures, and Eller et al. [52] reported that higher temperatures and CO_2_ concentrations positively impacted the reed growth and overall biomass. High temperatures and water level fluctuations may also have a synergetic effect, thus influencing common reed productivity. For example, longer periods of reed submergence in combination with higher temperatures may leave reeds vulnerable to fungal pathogens, as was reported at lake Constance [53], which, in turn, can cause lower biomass. It is also possible that higher temperatures increase the loss of nitrogen through increased denitrification due to the decreased solubility of oxygen in the water.

Interestingly, a lack of water in the summer months only explains 9.9% of the total variance. High average temperatures in the summer months also showed no effect on the observed parameters, which was unexpected since water shortage became most detrimental at high temperatures. Common reeds can tolerate periods of drought, but water deficits reduce the size of leaves, increase leaf shedding, and decreases the production of new leaves [54]. This is probably due to a decreased photosynthetic capacity and photoinhibition during the daytime [55]. These responses ultimately lower the total AGB of reeds. In addition, this may explain the biomass difference between the riparian and littoral reed stands, as the littoral reed stand is more often subjected to drought, and the impact of dry days was detected. However, high water levels still exerted a greater impact on reed productivity, confirming that the main limiting factor of reed growth and productivity at lake Cerknica was water level fluctuations [29]. At lake Balaton, different morphological ecotypes were determined in relation to water depth [56]. It is also possible that the observed reed stands at lake Cerknica contain different morphological ecotypes due to differing water-level characteristics, but studies of additional data related to morphology are necessary to confirm this.

Temperatures prior to the establishment of reeds explain 11.6% of the total variance, with average temperatures accounting for 7.4% of the variance. The period before the establishment of reeds mainly provided an estimate of the beginning of the growth of common reeds. The onset, end, and length of growing seasons are related to features of climate, such as temperature, humidity, and water availability [57]. In the temperate zone of the northern hemisphere, the temperature is thought to be the primary factor determining growing season traits [58,59]. In our study, higher temperatures before reed establishment had an impact on the start of the growing season reed. This was most apparent at the littoral reed stand, as the earliest initial growth date was recorded in 2018 when early temperatures were the highest in the observed 5-year period. A negative relationship between plant AGB and high temperatures before the vegetative phase was also detected. Flooding events during the vegetative phase had a negative impact on the final biomass, but the absence of flooding before reed establishment yielded similar results. This, again, was most apparent in the littoral reed stand. A probable consequence of the absence of flooding before reed establishment was lower soil moisture. Interactions between plants and soil moisture are pivotal in ecohydrology [60] and have shown that soil moisture directly impacts the establishment and growth of plants and leaf phenology [61]. Low levels of soil moisture can induce drought stress on vegetation, thus directly impacting production [62]. The riparian reed stand at Gorenje Jezero is less likely to be affected by this issue due to its proximity to the river. This confirms that water levels are the key factor driving common reed production at lake Cerknica. 

## 5. Conclusions

This study has revealed that the main factor driving the production and temporal dynamics of common reed at lake Cerknica is fluctuations in the water level. Data concerning the water level explicated a total of 62.5% of the variance. The use of an NDVI-derived growth model allowed an exploration of the effect of environmental variables, which are mainly water-level fluctuations, and temperatures in different growth stages from 2017 to 2021. A comparison of littoral and riparian reed stands, the main difference between which is the extent and duration of flooding, showed that intense and long flooding events during the vegetative phase negatively impacted the above-ground biomass production of common reeds. This was most apparent in 2019 when the water level peaked above 1.5 m during the vegetative phase at the littoral site and resulted in significantly lower biomass. In 2019, the AGB of the littoral reeds was 589 g lower than that of the riparian reed. The absence of floods before the reed establishment negatively impacted the littoral reed stand. High temperatures and a lack of flooding, indicating a drought during the summer months, also exerted a negative effect on the production of littoral reeds, but to a lesser degree. Constant and more moderate water availability at the riparian reed stand was shown to be more favorable for the growth and production of common reeds.

## Figures and Tables

**Figure 1 plants-12-01006-f001:**
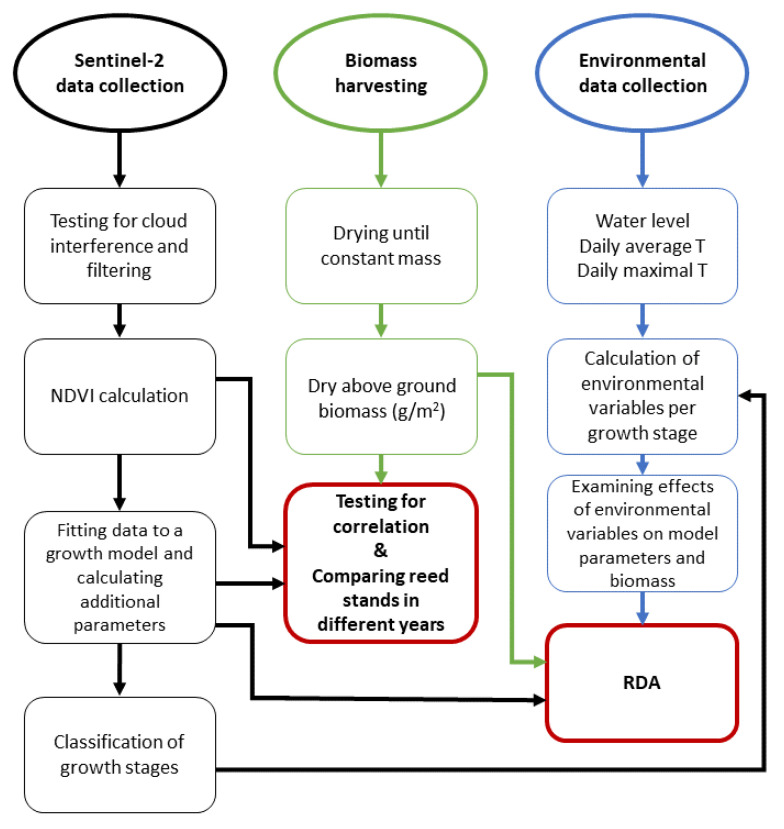
Workflow of this research. Red colored squares represent statistical analyses while the rest of the squares (black—remote sensing part, green—fieldwork part, blue—environmental data part) represent the steps that were adopted to achieve the results.

**Figure 2 plants-12-01006-f002:**
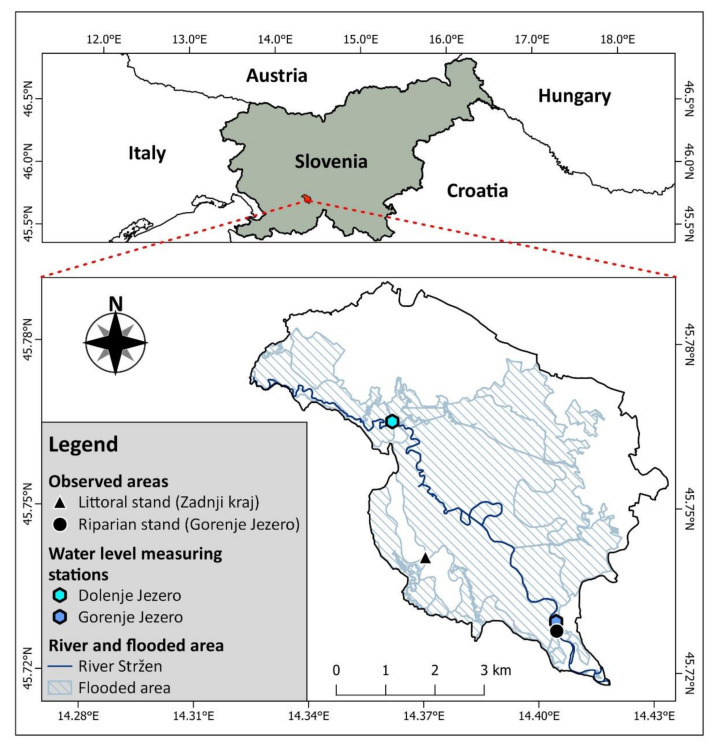
Map of lake Cerknica showing the observed areas of reed stands, the locations of water level measuring stations, floodplains, and the river Stržen.

**Figure 3 plants-12-01006-f003:**
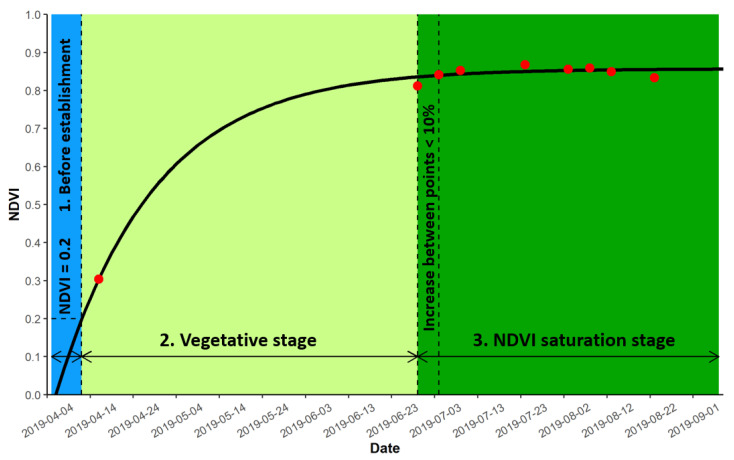
Visual representation of classifying three different stages based on the growing season, model, and NDVI dataset derived from Sentinel-2 images (red points).

**Figure 4 plants-12-01006-f004:**
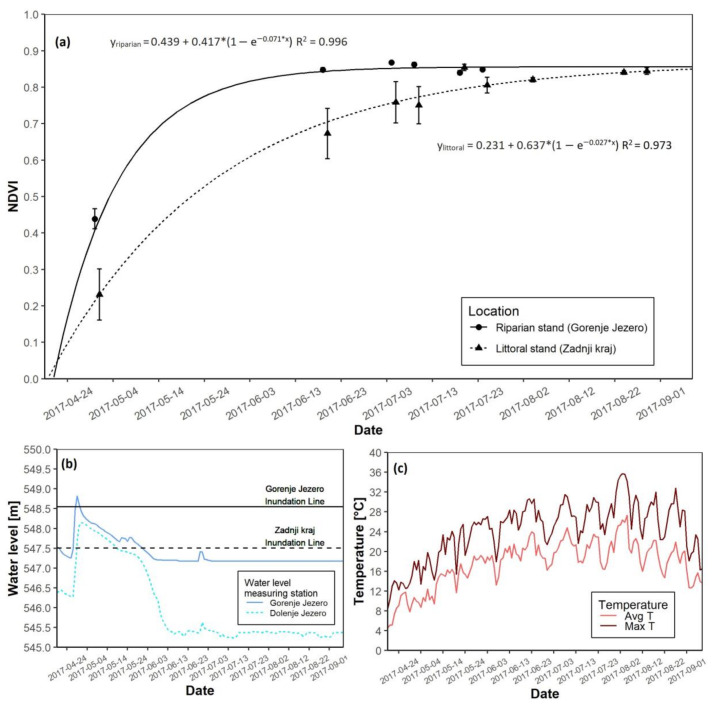
NDVI growth models for both reed sites with water level data for the 2017 growing season. (**a**) NDVI growth models for the riparian and littoral reed stands; (**b**) Water level data for the riparian and littoral reed stands; (**c**) Daily average and maximal temperatures.

**Figure 5 plants-12-01006-f005:**
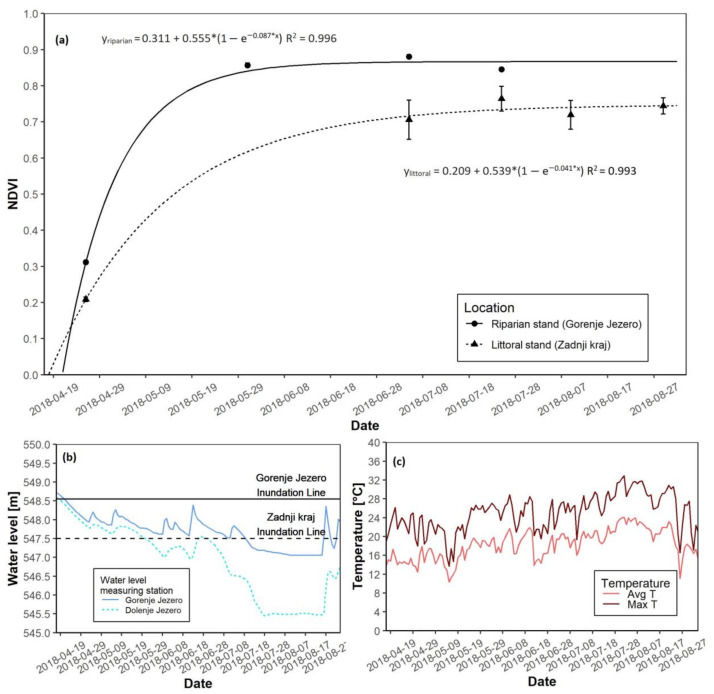
NDVI growth models for both reed sites with water level data for the 2018 growing season. (**a**) NDVI growth models for the riparian and littoral reed stands; (**b**) Water level data for the riparian and littoral reed stands; (**c**) Daily average and maximal temperatures.

**Figure 6 plants-12-01006-f006:**
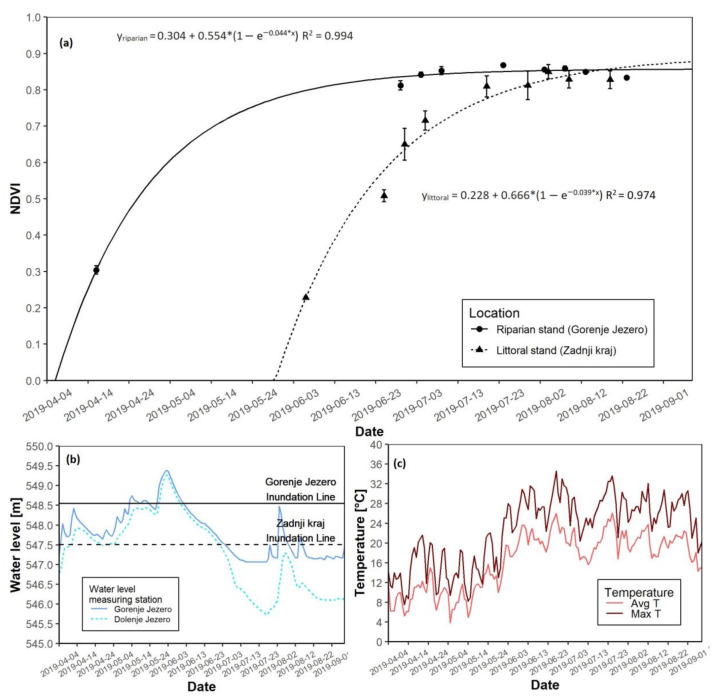
NDVI growth models for both reed sites with water level data for the 2019 growing season. (**a**) NDVI growth models for the riparian and littoral reed stands; (**b**) Water level data for the riparian and littoral reed stands; (**c**) Daily average and maximal temperatures.

**Figure 7 plants-12-01006-f007:**
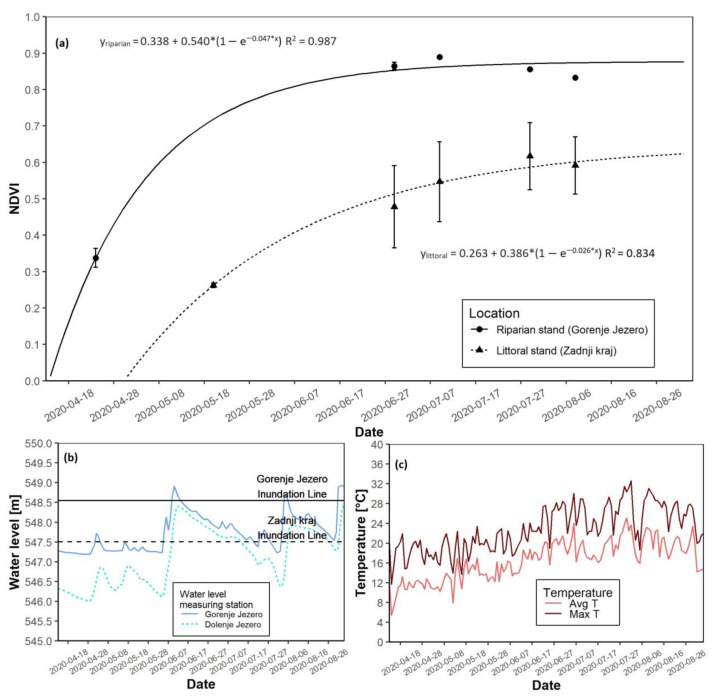
NDVI growth models for both reed sites with water level data for the 2020 growing season. (**a**) NDVI growth models for the riparian and littoral reed stands; (**b**) Water level data for the riparian and littoral reed stands; (**c**) Daily average and maximal temperatures.

**Figure 8 plants-12-01006-f008:**
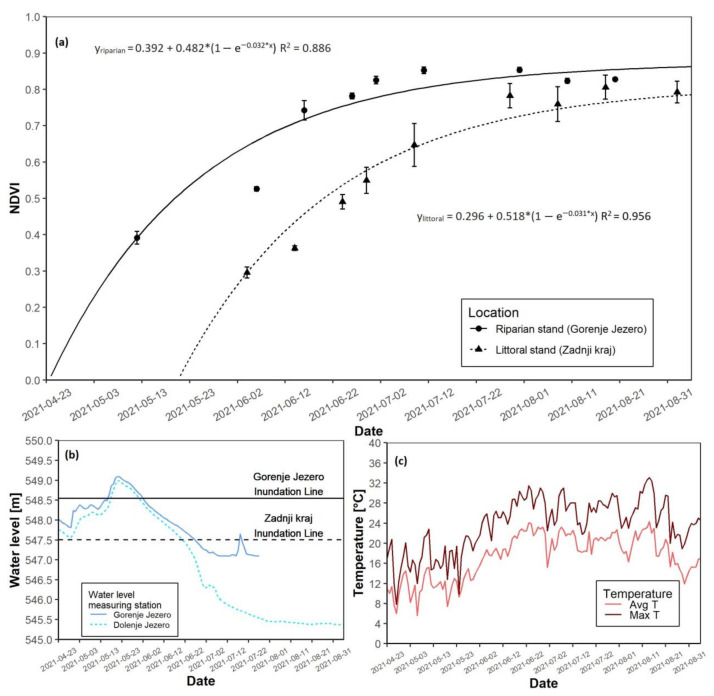
NDVI growth models for both reed sites with water level data for the 2021 growing season. (**a**) NDVI growth models for the riparian and littoral reed stands; (**b**) Water level data for the riparian and littoral reed stands; (**c**) Daily average and maximal temperatures. Data regarding water-level at Gorenje Jezero ends abruptly in July due to missing data.

**Figure 9 plants-12-01006-f009:**
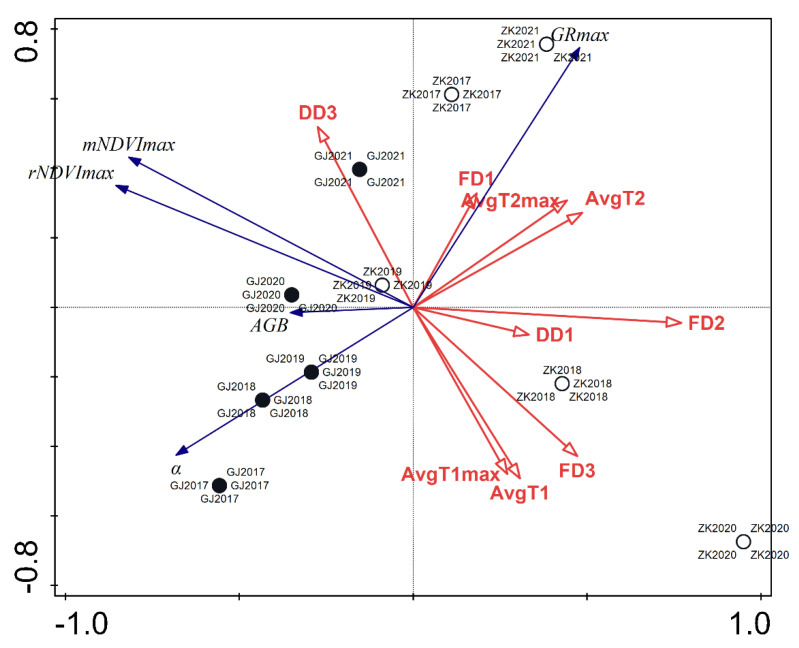
RDA triplot showing the relationship between model parameters and biomass with environmental variables from 2017 to 2021 for the littoral reed stand at Zadnji kraj and the riparian reed stand at Gorenje Jezero. Black and white circles indicate both locations depending on the year. Blue vectors indicate dependent variables and red vectors indicate independent variables. (GJ—Gorenje Jezero riparian stand; ZK—Zadnji kraj littoral stand; FD3—number of flooded days during NDVI saturation; FD2—number of flooded days during the vegetative phase; FD1—number of flooded days before reed establishment; DD3—number of dry days during the NDVI saturation period; DD1—number of dry days before reed establishment; AvgT1—average daily temperature before reed establishment; AvgT1_max_—average daily maximal temperature before reed establishment; AvgT2—average daily temperature during the vegetative phase; AvgT2_max_—average daily maximal tempderature during the vegetative phase; rNDVI_max_—maximal NDVI value of our NDVI dataset; mNDVI_max_—maximal NDVI value predicted by the model; GR_max_—time of maximal growth increase; *α*—initial rate of NDVI increase; AGB—above-ground biomass at the end of the growing season).

**Table 1 plants-12-01006-t001:** Types of data acquired from measuring stations, locations of measuring stations with latitudes and longitudes, and their corresponding reed stands.

Data	Measuring Station	Corresponding Stand
Water level data	Dolenje Jezero (45.765512, 14.361263)	Littoral stand (Zadnji kraj)
	Gorenje Jezero (45.728317, 14.404899)	Riparian stand (Gorenje Jezero)
Temperature data	Postojna (45.766049, 14.193119)	Both stands

**Table 2 plants-12-01006-t002:** Correlation results between AGB, modelled maximal NDVI (mNDVI_max_), and maximal NDVI values (rNDVI_max_). Spearman’s correlation test was used.

Correlation between Variables	*p*-Value	Correlation Coefficient
AGB and mNDVI_max_	0.875	0.026
AGB and rNDVI_max_	0.028	0.347

**Table 3 plants-12-01006-t003:** Model results and above-ground biomass at the end of each growing season from 2017 to 2021 for the riparian reed stand at Gorenje Jezero and the littoral reed stand at Zadnji kraj. (GS—the date of growth-start; rNDVI_max_—maximal NDVI value of our NDVI dataset; mNDVI_max_—maximal NDVI value predicted by model; GR_max_—date of maximal growth rate; α—initial rate of NDVI increase; AGB—above-ground biomass at the end of the growing season expressed in grams per square meter). Compact letter displays show differences between values based on the Dunn post hoc test.

Location	Season	GS	rNDVI_max_	mNDVI_max_	GR_max_	α	AGB (g/m^2^)
Riparian stand Gorenje Jezero	2017	2017-04-25	0.855 ^ab^ ± 0.003	0.856 ^abc^ ± 0.027	2017-05-05	0.217 ^a^ ± 0.032	594.3 ^ab^ ± 125.7
	2018	2018-04-24	0.880 ^a^ ± 0.005	0.866 ^abc^ ± 0.005	2018-05-02	0.136 ^ab^ ± 0.002	625.6 ^ab^ ± 102.5
	2019	2019-04-12	0.867 ^ab^ ± 0.004	0.857 ^abc^ ± 0.011	2019-04-29	0.067 ^abcd^ ± 0.002	804.3 ^a^ ± 241.2
	2020	2020-04-19	0.889 ^a^ ± 0.001	0.877 ^ab^ ± 0.025	2020-05-06	0.08 ^abc^ ± 0.007	761.4 ^ab^ ± 282.2
	2021	2021-05-02	0.853 ^abc^ ± 0.006	0.874 ^ab^ ± 0.017	2021-05-23	0.07 ^abc^ ±0.004	867.4 ^a^ ± 239.5
Littoral stand Zadnji kraj	2017	2017-04-28	0.854 ^abc^ ± 0.013	0.868 ^abc^ ± 0.07	2017-05-26	0.034 ^d^ ± 0.006	366.9 ^ab^ ± 29.8
	2018	2018-04-26	0.764 ^bc^ ± 0.034	0.747 ^c^ ± 0.006	2018-05-13	0.045 ^cd^ ± 0.001	390 ^ab^ ± 147.4
	2019	2019-06-05	0.849 ^abc^ ± 0.019	0.893 ^a^ ± 0.003	2019-06-24	0.05 ^bcd^ ± 0.001	215.3 ^b^ ± 36.4
	2020	2020-05-15	0.617 ^c^ ± 0.092	0.648 ^c^ ± 0.006	2020-06-07	0.035 ^d^ ± 0.001	476.5 ^ab^ ± 141.2
	2021	2021-05-30	0.805 ^bc^ ± 0.032	0.813 ^bc^ ± 0.015	2021-06-22	0.046 ^cd^ ± 0.002	534.6 ^ab^ ± 164.7

**Table 4 plants-12-01006-t004:** The numerical results (conditional effects) of RDA. Ratio of variance in model parameters and above-ground biomass at the end of the season explained by water level data and temperature data at both reed stands. (FD3—number of flooded days during NDVI saturation; FD2—number of flooded days during the vegetative phase; FD1—number of flooded days before reed establishment; DD3—number of dry days during NDVI saturation; DD1—number of dry days before reed establishment; AvgT1—average daily temperature before reed establishment; AvgTmax1—average daily maximal temperature before reed establishment; AvgT2—average daily temperature during the vegetative phase; AvgT2max—average daily maximal temperature during the vegetative phase). Significance of all explained variances = 0.002.

Group Variables	Variable	% of Explained Variance
Water level—flood	FD3	10.8
	FD2	28.6
	FD1	5.7
Water level—dry	DD3	9.9
	DD1	7.5
Temperatures	AvgT1	7.4
	AvgT1_max_	4.2
	AvgT2	5
	AvgT2_max_	8.8

## Data Availability

Data will be provided upon reasonable request.

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
