# Peer review of "Contrasting Dynamics of Littoral and Riparian Reed Stands within a Wetland Complex of Lake Cerknica"

_plants, 2023, doi:10.3390/plants12051006_

Round 1

Reviewer 1 Report

Thank you for the interesting paper. Please find my comments and suggestions from the attached manuscript.

Author Response

We would like to thank the reviewer for very relevant comments that helped us improve our manuscript. Below you will find answers and explanations related to each comment (referring to lines is based on the clean version of the manuscript).

Reviewer 1/A

English language and style

( ) English very difficult to understand/incomprehensible
( ) Extensive editing of English language and style required
( ) Moderate English changes required
(x) English language and style are fine/minor spell check required
( ) I don't feel qualified to judge about the English language and style

Yes

Can be improved

Must be improved

Not applicable

Does the introduction provide sufficient background and include all relevant references?

( )

(x)

( )

( )

Are all the cited references relevant to the research?

( )

(x)

( )

( )

Is the research design appropriate?

( )

( )

(x)

( )

Are the methods adequately described?

( )

( )

(x)

( )

Are the results clearly presented?

( )

(x)

( )

( )

Are the conclusions supported by the results?

( )

(x)

( )

( )

Comments and Suggestions for Authors

Thank you for the interesting paper. Please find my comments and suggestions from the attached manuscript.

A1

Comment: You need to address the issue of water quality before comparing the NDVI values of two different sites.

Response: Thank you for this comment. Water quality of both observed areas is deemed as comparable. This has been added to the text with references (391-392):

https://www.sciencedirect.com/science/article/abs/pii/0273122396002521

https://onlinelibrary.wiley.com/doi/full/10.1111/j.1440-1770.2003.00228.x

A2

Comment: Isn't this taken from NDVI value? Two models in Figure 1 shows that 0.2 NDVI value has different dates. Wasn't 0.2 NDVI considered date of growth start? Line 407

Response: Thank you for this comment. What we meant by this is that the first of May was the initial used NDVI value at both sites (NDVI acquired from Sentinel-2 images). We realise that the way we worded this causes confusion. We substituted initial used NDVI dates with dates where NDVI reached 0.2 (since we established this as growth start) with the abbreviation “GS” (table 2). Hopefully this improves clarity of the text. We also applied this in results.

A3

Comment: What are those parameters? Please describe them.

Response: Water level data and data regarding temperature. Wording has been changed from “parameters” to “variables”. They are described under “Environmental variables”.

A4

Comment: Does this mean only 8 pixels were analyzed for this study? Were those pixel dispersed or clumped? If dispersed, how were they selected?

If so, this is a too small a sample to draw any conclusion regarding the relationship between dry mass and NDVI and other analysis.

Response: Selected pixels were clumped for a combined observed area of 400m2. We refrained from using more pixels as the results could be misleading as there would be a higher chance that the extra pixels would have mixed vegetation. We clarified in text that the selected pixels are adjacent (420).

A5

Comment: “Data validation was performed on the site” What does this mean?

Response: Thank you for this comment. Data validation was a poor use of words. What we meant by this is that we verified on site that the selected pixels contained common reed as the dominant plant species. Mistake has been corrected.

A6

Comment: The NDVI values of two different sites could be affected by the difference of water quality (Presence of total suspended solid). Are you assuming that the water quality of the riparian and littoral area is similar?

Response: Thank you for this concern. Quality of water between both locations is similar. When the lake is flooded the effect of tributaries can be neglected and common reed along with other macrophytes act as a buffer against pollution. Eutrophication processes are also slowed down due to fast mineralization during dry periods and reed decline (common effect of eutrophication) has never been reported at these sites. Two additional references have been added that clarify the issue of water quality of the observed sites in the “Study area” paragraph.

A7

Comment: How many models?

Response: Thank you for this comment. Sentence was worded poorly. What we meant was that we fit the data for each year and location to the same model. Sentence has been changed accordingly.

A8

Comment: Is this one plot each pixel? Please clarify.

Response: 4 plots were harvested within the 400m2 stand, not necessarily within each pixel. Even though this is a protected area, farmers owning land at lake Cerknica often harvest common reed. Extent of harvesting changes yearly as it depends on ground characteristics connected with flooding/drying. We harvest our plots yearly where we can within the 400m2 areas. This is also the reason why senescence part of phenology was ignored as reed harvesting would impact remotely sensed data.

A9

Comment: Please list them and describe them briefly.

Response: Variables have been listed. They are also described under Environmental variables in methods.

A10

Comment:  Are these the environmental parameters used for the study?

Response: We described used variables in the added part of the text describing the RDA.

A11

Comment: Please describe the analysis briefly as it appears to be the main analysis method.

Response: A sentence was added describing input data for the redundancy analysis and its purpose which adds additional value to the current description (483-486).

Reviewer 2 Report

The manuscript with the title “Different dynamics of littoral and riparian reed stands within a wetland complex of lake Cerknica” is an attempt to use NDVI derived from Sentinel-2 images and field data to analyse the impact of different meteorological parameters on littoral and riparian reed’s NDVI. While the idea might be useful, the manuscript contains many drawbacks that need to be addressed before audiences/reviewers can read it clearly. I strongly advise the authors revised the manuscript before resubmitting it. 

-          The first problem is the abstract. Authors need to address why we need to conduct the research (the importance of the research), but the abstract is lack of it.

-          The introduction needs to be revised as well. While authors barely mentioned about the Common reed’s importance and the use of remote-sensed data to study them, there is no clear aim of this study. What they described (L68-76) are the methodology of this research, not the aim and objectives.

-          The other problem is the paper’s structure. They put the methodology at the end of the paper while audiences do not know how they conduct their research. It is a strange structure for a scientific manuscript.

-          The methodology needs to be improved as well. It is very hard to read and follow information provided in the methodology section. For example

o Authors mentioned two stations where they collected environmental parameters, they should provide information as a table for better clarification (L347-353).

o   Link to the sources of Sentinel-2 images should be provided (L361-363)

o   “Each image was tested for cloud interference and inappropriate images were filtered out.” (L365-366). Please specify this, it is too general.

o   What is “Data validation was performed on the site”? (L367-368) How could you do this?

o   L375-376: What is “NDVI saturation”? You need to define this because it has several meanings for the same term.

o   L406-411: An additional graph/diagram is a better choice to explain this for audiences

Author Response

We would like to thank the reviewer for very relevant comments that helped us improve our manuscript. Below you will find answers and explanations related to each comment (referring to lines is based on the clean version of the manuscript).

Reviewer 2 / B

English language and style

( ) English very difficult to understand/incomprehensible
( ) Extensive editing of English language and style required
( ) Moderate English changes required
(x) English language and style are fine/minor spell check required
( ) I don't feel qualified to judge about the English language and style

Yes

Can be improved

Must be improved

Not applicable

Does the introduction provide sufficient background and include all relevant references?

( )

( )

(x)

( )

Are all the cited references relevant to the research?

( )

(x)

( )

( )

Is the research design appropriate?

( )

(x)

( )

( )

Are the methods adequately described?

( )

( )

(x)

( )

Are the results clearly presented?

( )

(x)

( )

( )

Are the conclusions supported by the results?

( )

( )

( )

(x)

Comments and Suggestions for Authors

The manuscript with the title “Different dynamics of littoral and riparian reed stands within a wetland complex of lake Cerknica” is an attempt to use NDVI derived from Sentinel-2 images and field data to analyse the impact of different meteorological parameters on littoral and riparian reed’s NDVI. While the idea might be useful, the manuscript contains many drawbacks that need to be addressed before audiences/reviewers can read it clearly. I strongly advise the authors revised the manuscript before resubmitting it. 

B1

Comment: The first problem is the abstract. Authors need to address why we need to conduct the research (the importance of the research), but the abstract is lack of it.

Response: Thank you for this comment. We improved the abstract by adding the sentence about the importance of our research and removing some general statements from the abstract.

B2

Comment: The introduction needs to be revised as well. While authors barely mentioned about the Common reed’s importance and the use of remote-sensed data to study them, there is no clear aim of this study. What they described (L68-76) are the methodology of this research, not the aim and objectives.

Response: Thank you for this comment. We removed statements that describe the methodology in the last paragraph of the introduction. Aim and objectives have been added in their place.

B3

Comment: The other problem is the paper’s structure. They put the methodology at the end of the paper while audiences do not know how they conduct their research. It is a strange structure for a scientific manuscript.

Response: We used the template suggested by the journal Plants and the structure of the paper therein. We followed the instruction for authors created by the Editorial office.

B4

Comment: The methodology needs to be improved as well. It is very hard to read and follow information provided in the methodology section. For example: Authors mentioned two stations where they collected environmental parameters, they should provide information as a table for better clarification (L347-353).

Response: Thank you for this comment. We inserted a table depicting which measuring station was used to gather temperature/water level data for each reed stand (line 408).

B5

Comment: Link to the sources of Sentinel-2 images should be provided (L361-363)

Response: We provided the link where the used images were obtained (https://scihub.copernicus.eu/dhus/#/home). Line 413

B6

Comment: “Each image was tested for cloud interference and inappropriate images were filtered out.” (L365-366). Please specify this, it is too general.

Response: Thank you for this comment. Cloud interference was tested using ESA’s SNAP software by creating a cloud mask for every image (IdePix S2-MSI plugin). If the cloud mask was layered over our observed sites, the corresponding Sentinel2 image was not used. Description has been added to the manuscript. (417-419).

B7

Comment: What is “Data validation was performed on the site”? (L367-368) How could you do this?

Response: Thank you for this comment. Data validation was a poor use of words. What we meant by this is that we verified on site that the selected pixels contained common reed as the dominant plant species. 

B8

Comment: L375-376: What is “NDVI saturation”? You need to define this because it has several meanings for the same term.

Response: Thank you for this comment. Sentence has been rewritten to describe which portion of the data was removed more accurately. (429-430).

B9

Comment: L406-411: An additional graph/diagram is a better choice to explain this for audiences

Response: Thank you for this comment. Visual representation of the three determined stages has been added (472).

Reviewer 3 Report

The work entitled "Different dynamics of littoral and riparian reed stands within a wetland complex of lake Cerknica" is interesting and insightful. However, there are suggestion written below:

1. Remove general statements from the abstract.

2. Add some more relevant and recent literatures into introduction.

3. Restructure paper as per the format: 1. Introduction, 2. Study Area, 3. Materials and Methods, 4. Results, 5. Discussion, 6. Conclusion

4. Authors may include field validation to strengthen the research.

5. In fig. 7, texts are not visible, latitude and longitude are also not clearly visible, write in degree decimal format

6. Conclusion needs little bit improvement in the term of data values.

Author Response

We would like to thank the reviewer for very relevant comments that helped us improve our manuscript. Below you will find answers and explanations related to each comment.

Reviewer 3 / C

Open Review

English language and style

( ) English very difficult to understand/incomprehensible
( ) Extensive editing of English language and style required
( ) Moderate English changes required
(x) English language and style are fine/minor spell check required
( ) I don't feel qualified to judge about the English language and style

Yes

Can be improved

Must be improved

Not applicable

Does the introduction provide sufficient background and include all relevant references?

( )

(x)

( )

( )

Are all the cited references relevant to the research?

( )

(x)

( )

( )

Is the research design appropriate?

(x)

( )

( )

( )

Are the methods adequately described?

(x)

( )

( )

( )

Are the results clearly presented?

(x)

( )

( )

( )

Are the conclusions supported by the results?

( )

(x)

( )

( )

Comments and Suggestions for Authors

The work entitled "Different dynamics of littoral and riparian reed stands within a wetland complex of lake Cerknica" is interesting and insightful. However, there are suggestion written below:

C1

Comment: Remove general statements from the abstract.

Response: Thank you for this comment. General statements were removed from the abstract and the importance of research was added in their place.

C2

Comment: Add some more relevant and recent literatures into introduction.

Response: Recent literature sources have been added.

C3

Comment: Restructure paper as per the format: 1. Introduction, 2. Study Area, 3. Materials and Methods, 4. Results, 5. Discussion, 6. Conclusion

Response: We used the template suggested by the journal Plants and the structure of the paper therein. We followed the instruction for authors created by the Editorial office.

C4

Comment: Authors may include field validation to strengthen the research.

Response: Thank you for this comment. Field validation of results consisted of dry above ground biomass per square meter. This was our main productivity parameter and other data was not used.

C5

Comment: In fig. 7, texts are not visible, latitude and longitude are also not clearly visible, write in degree decimal format

Response: Thank you for this comment. Text has been resized, latitude and longitude has been converted to degree decimal format. A map of Slovenia has also been added with highlighted lake Cerknica.

C6

Comment: Conclusion needs little bit improvement in the term of data values.

Response: Data values have been added to Conclusions.

Reviewer 4 Report

It is a good work, with an adequate design. But it has flaws that need to be fixed before it can be resubmitted for publication.   The file with specific revisions is attached.

In a second version, the authors must indicate how they resolved each observation.

Author Response

We would like to thank the reviewer for very relevant comments that helped us improve our manuscript. Below you will find answers and explanations related to each comment (referring to lines is based on the clean version of the manuscript).

Reviewer 4 / D

English language and style

( ) English very difficult to understand/incomprehensible
( ) Extensive editing of English language and style required
( ) Moderate English changes required
(x) English language and style are fine/minor spell check required
( ) I don't feel qualified to judge about the English language and style

Yes

Can be improved

Must be improved

Not applicable

Does the introduction provide sufficient background and include all relevant references?

( )

( )

(x)

( )

Are all the cited references relevant to the research?

( )

( )

(x)

( )

Is the research design appropriate?

(x)

( )

( )

( )

Are the methods adequately described?

( )

( )

(x)

( )

Are the results clearly presented?

( )

( )

(x)

( )

Are the conclusions supported by the results?

( )

(x)

( )

( )

Comments and Suggestions for Authors

It is a good work, with an adequate design. But it has flaws that need to be fixed before it can be resubmitted for publication.   The file with specific revisions is attached.

In a second version, the authors must indicate how they resolved each observation

D1

Comment: Please rewrite the Introduction in such a way that the ideas are expressed in a logical sequence. Paragraphs are disconnected from each other.

Response: Thank you for this comment. We changed the position of certain paragraphs and added text to connect them in a logical manner.

D2

Comment: It appears that you misinterpreted the results of Dunn’s test. For example, in Riparian stand, in the column rNDVImax, all figures share the letter “a”; so, there are no differences between seasons. This problem is repeated in mNDVImax, and α. Also in α from Littoral stand.

In Littoral stand doesn’t make sense that 0.747 had “a”, but 0.893, that is bigger, had “b”. It is also very important that you include in the results all the statistics obtained with their respective probabilities of rejecting or not the null hypotheses.

Response: Thank you for these comments and remarks. Comparisons were conducted between all years and both locations. For example, mean rNDVImax from the riparian stand in year 2017 was compared with all other years at the same location and with the littoral reed stand (same comparison was done for every location x season pair). That is why the compact letter display shows more than one letter at the riparian site even though there were no significant differences between years. However, significant differences were observed when comparing the riparian reed stand and littoral reed stand in different years.

We are not sure what is wrong with the value 0.747 having the cld as “a” while value 0.893 has the cld as “b”. p-value of the comparison is 0.022738102 indicating a significant difference. Would it be more correct for 0.893 to have letter “a” as it is the higher value of the two? If that is indeed the case, we corrected this by switching the two compact letter displays.

Regarding including probabilities of rejecting/accepting the null hypothesis. We are unable to include all the p-values as there is simply too many comparisons. That is why compact letter displays are added to display where significant differences were reported. For example, here are the results of comparisons of just the mNDVImax parameter (We have also attached a file with Responses):

        Comparison           Z      P.unadj       P.adj

1  GJ2017 - GJ2018 -0.36291503 7.166684e-01 1.000000000

2  GJ2017 - GJ2019  0.09072876 9.277081e-01 1.000000000

3  GJ2018 - GJ2019  0.45364378 6.500852e-01 1.000000000

4  GJ2017 - GJ2020 -0.72583005 4.679430e-01 1.000000000

5  GJ2018 - GJ2020 -0.36291503 7.166684e-01 1.000000000

6  GJ2019 - GJ2020 -0.81655881 4.141806e-01 1.000000000

7  GJ2017 - GJ2021 -0.66534422 5.058304e-01 1.000000000

8  GJ2018 - GJ2021 -0.30242919 7.623249e-01 1.000000000

9  GJ2019 - GJ2021 -0.75607297 4.496054e-01 1.000000000

10 GJ2020 - GJ2021  0.06048584 9.517687e-01 1.000000000

11 GJ2017 - ZK2017 -0.54437254 5.861851e-01 1.000000000

12 GJ2018 - ZK2017 -0.18145751 8.560085e-01 1.000000000

13 GJ2019 - ZK2017 -0.63510130 5.253624e-01 1.000000000

14 GJ2020 - ZK2017  0.18145751 8.560085e-01 1.000000000

15 GJ2021 - ZK2017  0.12097168 9.037135e-01 1.000000000

16 GJ2017 - ZK2018  1.84481806 6.506401e-02 1.000000000

17 GJ2018 - ZK2018  2.20773308 2.726289e-02 1.000000000

18 GJ2019 - ZK2018  1.75408930 7.941521e-02 1.000000000

19 GJ2020 - ZK2018  2.57064811 1.015084e-02 0.456787849

20 GJ2021 - ZK2018  2.51016227 1.206757e-02 0.543040619

21 ZK2017 - ZK2018  2.38919060 1.688554e-02 0.759849294

22 GJ2017 - ZK2019 -1.63311762 1.024443e-01 1.000000000

23 GJ2018 - ZK2019 -1.27020260 2.040125e-01 1.000000000

24 GJ2019 - ZK2019 -1.72384638 8.473558e-02 1.000000000

25 GJ2020 - ZK2019 -0.90728757 3.642547e-01 1.000000000

26 GJ2021 - ZK2019 -0.96777341 3.331575e-01 1.000000000

27 ZK2017 - ZK2019 -1.08874508 2.762663e-01 1.000000000

28 ZK2018 - ZK2019 -3.47793568 5.052912e-04 0.022738102

29 GJ2017 - ZK2020  2.32870476 1.987471e-02 0.894362033

30 GJ2018 - ZK2020  2.69161979 7.110596e-03 0.319976811

31 GJ2019 - ZK2020  2.23797600 2.522262e-02 1.000000000

32 GJ2020 - ZK2020  3.05453481 2.254099e-03 0.101434433

33 GJ2021 - ZK2020  2.99404898 2.753017e-03 0.123885787

34 ZK2017 - ZK2020  2.87307730 4.064946e-03 0.182922589

35 ZK2018 - ZK2020  0.48388670 6.284663e-01 1.000000000

36 ZK2019 - ZK2020  3.96182238 7.437987e-05 0.003347094

37 GJ2017 - ZK2021  1.17947384 2.382095e-01 1.000000000

38 GJ2018 - ZK2021  1.54238887 1.229791e-01 1.000000000

39 GJ2019 - ZK2021  1.08874508 2.762663e-01 1.000000000

40 GJ2020 - ZK2021  1.90530389 5.674058e-02 1.000000000

41 GJ2021 - ZK2021  1.84481806 6.506401e-02 1.000000000

42 ZK2017 - ZK2021  1.72384638 8.473558e-02 1.000000000

43 ZK2018 - ZK2021 -0.66534422 5.058304e-01 1.000000000

44 ZK2019 - ZK2021  2.81259146 4.914405e-03 0.221148203

45 ZK2020 - ZK2021 -1.14923092 2.504608e-01 1.000000000

D3

Comment: Please include the units. Grams per square meter?

Response: Thank you for this comment. Units were expressed as grams per plot. In order for our results to be comparable with similar research, we expressed AGB as g/m2.

D4

Comment: Please include the equations of the two models (equations of the curves), as well as the statistic and level of significance. Do the same in figures 2-5 and in the appendices.

Response: Thank you for this comment. Equations have been added along with their R-squared for all figures and supplementary material. Table S1 (table with all equations with R-squared values and p-values of constants) has been added under supplementary material.

D5

Comment: Check if you are using the term "parameter" correctly, since the origin of this word comes from parametric statistics. Not all variables behave like parameters.

Response: Thank you for this comment. Wording “parameter” has been replaced with “variable” where it was applicable.

D6

Comment: Delete this column. If all the variables had the same significance, indicate it in the legend

Response: Column has been deleted with added text regarding significance value in table description.

D7

Comment: Replace "et." with "et". This is a recurring error in the text. Please correct where necessary.

Response: The error has been corrected throughout the text.

D8

Comment: Add a map of the country, another of the basin, locating it in the country, and locate the wetland complex within the basin.

Response: Thank you for this comment. A map depicting the location of lake Cerknica within Slovenia has been added. Map of the basin was not included as the area of lake Cerknica already covers the vast majority of the basin.

D9

Comment: Please use italics in all the components of the formulas to better understand the reading.

Response: Thank you for this comment. All components of the formulas have been italicized to improve readability.

D10

Comment: Please explain why this procedure allows you to get a better fit. What in particular does it have to use 51 and 25 days regarding another period of days.

Response: A major component of the model is the y0 variable (initial NDVI from the dataset generated from the Sentinel-2 satellite). In years 2019 and 2020 we had large gaps of missing data at the start of the growing season at the littoral site due to cloud coverage. Our initial NDVI values for those years are valued close to 0.2 (value indicating dead vegetation and bare soil) which is ok, however due to missing data we have no way of knowing if the value increases or stays the same in the following 5,10,15… days (Sentinel-2 has a revisit time of 5 days). Due to very successful fits in years where missing data was not an issue, we decided to move the initial NDVI value to a different date. We wrote a function that moved the date of the initial NDVI value by one day forward. We simply ran the function until we reached a significant fit. The date transformations serve as predictions or rather estimations of when reed growth started for years where cloud coverage made it impossible to get these NDVI values. Without these transformations, data cannot be fit to the model as the missing data gap underestimates the date of y0 and results in an awkward fit.

D11

Comment: Please justify this sample size and plot size. Common reed is a plant of considerable height, so large plots help reduce variability. Small plots are recommended when the plants are small.

Response: Common reed biomass has been the object of numerous studies at lake Cerknica. Plot size of 0.25m2 has always been the standard for harvesting biomass at this location. Articles that support this:

https://www.sciencedirect.com/science/article/pii/S1476945X09000890

https://link.springer.com/article/10.1007/s10750-015-2492-x

https://www.mdpi.com/2073-4441/12/10/2806

https://www.sciencedirect.com/science/article/pii/S0304377010000471

D12

Comment: Lines 421-423 (DCA):

When there is an unimodal response, it is assumed to be a broad gradient and non-parametric statistics are recommended. On the contrary, when a linear relationship is obtained, it is assumed that the gradient is short and parametric statistics are used. Include in the results the length of the gradient and how they proceeded based on this result.

Response: Thank you for this comment. An error was conducted on our part. When we imported datasets regarding explanatory (environmental variables) and explored (biomass, parameters acquired from remote sensing) values, we imported them as constitutional data. That is the reason why the next step was the DCA which resulted in a gradient of 0.15 (low enough to approach a linear response). However, our data is not constitutional which means that a unimodal response is not possible anyway and the only possible analysis is the linear RDA. We corrected this by removing the sentence about DCA as it was not needed in the first place. (481-489).

D13

Comment: Do not include appendices in the main text. Move them to the supplementary material.

Response: Appendices have been removed from the main text and added to a separate file.

D14

Comment: x-axis .Please correct in all cases where necessary.

Response: We corrected this mistake in all cases.

D15

Comment: Do not use capital letters in each word of the title. Review the instructions for authors. Correct in all cases that are necessary.

Response: Correction has been applied.

D16

Comment: All scientific names should be italicized. The genus name starts with a capital letter; while the specific epithet with lower case.

Correct in all cases that are necessary.

Response: Correction has been applied.

D17

Comment: Abbreviated Journal Name.

Correct it in all cases that is necessary.

Response: Corrected in all cases.

D18

Comment: Complete the reference

Response: Reference has been completed.

Reviewer 5 Report

Brief comments:

Recommend changing title to read as follows:

Contrasting dynamics of littoral and riparian reed stands within a  wetland complex of lake Cerknica, Slovenia.

Please add Phragmites australis to key words, since the scientific name is not in the title.

Author Response

Reviewer 5 / E

English language and style

( ) English very difficult to understand/incomprehensible
( ) Extensive editing of English language and style required
( ) Moderate English changes required
(x) English language and style are fine/minor spell check required
( ) I don't feel qualified to judge about the English language and style

Yes

Can be improved

Must be improved

Not applicable

Does the introduction provide sufficient background and include all relevant references?

(x)

( )

( )

( )

Are all the cited references relevant to the research?

(x)

( )

( )

( )

Is the research design appropriate?

(x)

( )

( )

( )

Are the methods adequately described?

( )

(x)

( )

( )

Are the results clearly presented?

( )

(x)

( )

( )

Are the conclusions supported by the results?

(x)

( )

( )

( )

Comments and Suggestions for Authors

E1

Comment: Recommend changing title to read as follows: Contrasting dynamics of littoral and riparian reed stands within a  wetland complex of lake Cerknica, Slovenia.

Response: Thank you for the suggestion, title has been changed.

E2

Comment: Please add Phragmites australis to key words, since the scientific name is not in the title.

Response: Scientific name has been added.

Round 2

Reviewer 2 Report

I believe the authors were well addressed and explained the problems I mentioned in the previous revision round. The only minor issue I think author need to revise is the table 4 (L408-409). Authors should provide the latitudes and longitudes of the stations along with other information. Aside from that, the manuscript is excellent for the publication

Author Response

Yes

Can be improved

Must be improved

Not applicable

Does the introduction provide sufficient background and include all relevant references?

(x)

( )

( )

( )

Are all the cited references relevant to the research?

(x)

( )

( )

( )

Is the research design appropriate?

(x)

( )

( )

( )

Are the methods adequately described?

(x)

( )

( )

( )

Are the results clearly presented?

(x)

( )

( )

( )

Are the conclusions supported by the results?

(x)

( )

( )

( )

Comments and Suggestions for Authors

B1

Comment: I believe the authors were well addressed and explained the problems I mentioned in the previous revision round. The only minor issue I think author need to revise is the table 4 (L408-409). Authors should provide the latitudes and longitudes of the stations along with other information. Aside from that, the manuscript is excellent for the publication

Response: Thank you for this comment, latitudes and longitudes have been added to the stations in table 4.

Reviewer 4 Report

The authors improved their work based on the reviewers’ comments. However, it is still necessary to present the results of Dunn's test correctly and make the small changes indicated in the attached files.

Author Response

Yes

Can be improved

Must be improved

Not applicable

Does the introduction provide sufficient background and include all relevant references?

(x)

( )

( )

( )

Are all the cited references relevant to the research?

(x)

( )

( )

( )

Is the research design appropriate?

(x)

( )

( )

( )

Are the methods adequately described?

(x)

( )

( )

( )

Are the results clearly presented?

(x)

( )

( )

( )

Are the conclusions supported by the results?

(x)

( )

( )

( )

Comments and Suggestions for Authors

The authors improved their work based on the reviewers’ comments. However, it is still necessary to present the results of Dunn's test correctly and make the small changes indicated in the attached files.

D1
Comment: I am attaching an example of how to interpret the results of Dunn's test. Please correct the data in the table according to that example, and then verify if the results changed. Then make the corresponding discussion.

Response: Thank you for this comment. Results were interpreted as per given example. Bellow you will find an example of the process we adopted to acquire compact letter displays. Comparisons gave only slightly different results, but the differences were negligible and did not require a change in discussion or description of the results.

D2

Comment: You should keep in mind that the replica should not be just to convince the reviewer. The same concerns that the reviewer has will surely arise in other readers, so be sure to write your manuscript in such a way that these concerns do not arise. Ideally, you should convince both the reviewer and the readers.

Response: Thank you for this comment. A sentence briefly explaining why the transformations were needed has been added.

D3

Comment: Same comment: you must convince both the reviewer and the readers. Therefore, this information should be included in the manuscript as well.

Response: Thank you for this comment, we added a sentence clarifying the plot sizes with a cited source.
